# West Nile and Usutu Viruses’ Surveillance in Birds of the Province of Ferrara, Italy, from 2015 to 2019

**DOI:** 10.3390/v13071367

**Published:** 2021-07-14

**Authors:** Alessandra Lauriano, Arianna Rossi, Giorgio Galletti, Gabriele Casadei, Annalisa Santi, Silva Rubini, Elena Carra, Davide Lelli, Mattia Calzolari, Marco Tamba

**Affiliations:** 1Epidemiology Unit, Istituto Zooprofilattico Sperimentale della Lombardia e dell’Emilia Romagna, 25124 Brescia, Italy; alessandra.lauriano@izsler.it (A.L.); arianna.rossi@izsler.it (A.R.); giorgio.galletti@izsler.it (G.G.); gabriele.casadei@izsler.it (G.C.); annalisa.santi@izsler.it (A.S.); 2Department of Emilia-Romagna Region, Istituto Zooprofilattico Sperimentale della Lombardia e dell’Emilia Romagna, 25124 Brescia, Italy; silva.rubini@izsler.it (S.R.); elena.carra@izsler.it (E.C.); mattia.calzolari@izsler.it (M.C.); 3Department of Animal Health, Istituto Zooprofilattico Sperimentale della Lombardia e dell’Emilia Romagna, 25124 Brescia, Italy; davide.lelli@izsler.it

**Keywords:** Usutu virus, West Nile virus, passive surveillance, wild birds

## Abstract

West Nile (WNV) and Usutu (USUV) viruses are mosquito-borne flaviviruses. Thanks to their importance as zoonotic diseases, a regional plan for surveillance of Arboviruses was implemented in Emilia-Romagna in 2009. The province of Ferrara belongs to the Emilia-Romagna region, and it is an endemic territory for these viruses, with favorable ecological conditions for abundance of mosquitoes and wild birds. From 2015 to 2019, we collected 1842 dead-found birds at a wildlife rehabilitation center, which were analysed by three different PCRs for the detection of WNV and USUV genomes. August was characterized by the highest infection rate for both viruses. Columbiformes scored the highest USUV prevalence (8%), while Galliformes and Strigiformes reported the highest prevalence for WNV (13%). Among Passeriformes (the most populated Order), *Turdus merula* was the most abundant species and scored the highest prevalence for both viruses. To optimize passive surveillance plans, monitoring should be focused on the summer and towards the avian species more prone to infection by both viruses.

## 1. Introduction

Usutu (USUV) and West Nile (WNV) viruses are neurotropic mosquito-borne flaviviruses, members of the Japanese encephalitis antigenic group. They were first discovered in Africa during the XX century, and they are now considered endemic in several European countries. USUV and WNV have probably the same life cycles, including ornithophylic mosquitoes (mainly *Culex* spp.) as the main vectors and wild birds as amplifying hosts. Mammals, like horses and humans, are accidental, dead-end hosts [1]. Stationary and migratory wild birds play a major role in the spreading and maintenance of the viruses [1]. As USUV and WNV are zoonotic viruses, surveillance programs are necessary for early detection of their circulation [2].

The province of Ferrara within the Italian region Emilia-Romagna, on the Adriatic Sea, includes one of the largest European wetland areas and is an endemic area for both USUV and WNV. Thanks to its geographical characteristics and the co-existence of mosquitoes, as well as stationary and migratory wild birds, this area is an optimal environment for epidemiological studies in order to optimize USUV and WNV surveillance programs [3,4,5]. Here, we present the results of a 5-year passive surveillance program covering wild birds in this province.

## 2. Materials and Methods

### 2.1. Survey Area

The survey was carried out in the province of Ferrara. This is an area of 2635 square kilometers with a population of ≈344,000 people. One-third of this population resides in the city of Ferrara. Compared with the average Italian population density (200 inhabitants/sq km), the density in this province is rather low (326 inhabitants/sq km in the city and 50 inhabitants/sq km in the rest of the province). The province of Ferrara is totally plain and includes one of the largest wetlands of Europe (the Po river delta) where mosquitoes and wild birds coexist, and where USUV and WNV have both been endemic for several years [5]. In this province, there are also about 2000 equids, which are distributed over approximately 500 farms.

### 2.2. Sample Collection

The province of Ferrara belongs to the Emilia-Romagna region, where an integrated regional plan for surveillance of WNV was implemented in 2009, extended to USUV in 2011, and today is part of the national plan for surveillance and control of Arboviruses that started in 2015 [6].

This surveillance plan includes the fortnightly capture of mosquitoes using a network of fixed traps, the surveillance of birds through their capture (active surveillance) and through the collection of dead-found birds (passive surveillance), the passive surveillance of cases of neurological disease in horses and by testing all viral meningoencephalitis cases in humans [7].

The birds used in this study were collected by common people and carried to the wildlife recovery center (CRAS) of Ferrara in order to give them first aid care.

In agreement with the competent Official Veterinary Service, the animals that died during the recovery were stored frozen and bi-weekly sent to the same laboratory for molecular testing.

The data recorded are the following: species, date and area of collection, and date of death. For the extent of this work, only birds collected from 2015 to 2019 were considered.

### 2.3. Samples Analyses

All birds collected were tested for both viruses. Organ samples (heart, brain, kidney, and spleen) from each bird were pooled, mechanically homogenized, and tested by real time PCRs to detect WNV [8,9,10] and USUV [11] RNAs. Samples tested positive were further submitted to traditional PCRs to obtain amplicons for sequencing. All positive samples were submitted to a Pan-flavivirus protocol targeting the NS5 gene [12], and to two specific protocols directed to the gene E of WNV [13] and USUV [14].

The obtained sequences were used to confirm the identity of detected viruses through BLAST search (https://blast.ncbi.nlm.nih.gov/Blast.cgi, accessed on 4 April 2020). Sequences were then aligned and inspected to detect mutations and to obtain the consensus sequence. Alignments were used to obtain pairwise and overall p-distance (pairwise deletion option). These analyses were performed with the MEGAx software [15].

In order to attempt virus isolation, the remaining part of organ homogenates of PCR-positive birds was inoculated in confluent monolayers of VERO (African green monkey) cells. Cells were incubated at 37 °C with 5% CO_2_ for seven days and observed daily to assess any cytopathic effect (CPEs). For WNV and USUV identification, supernatant fluids of inoculated cell cultures were also tested through three different mAb-based sandwich ELISA specifically reactive against WNV and USUV or cross-reactive among members of Japanese encephalitis antigenic complex [16]. In the absence of CPEs’ and ELISAs’ reactivity, the cryolysates were sub-cultured twice onto fresh monolayers.

## 3. Results

From January 2015 to December 2019, 1842 dead-found birds, belonging to 17 orders and 82 species, were collected and analysed by PCR to detect the presence of USUV and/or WNV (Table 1, Appendix A).

Passeriformes was the most abundant order (794 samples), followed by Columbiformes (345 samples), Falconiformes (150 samples), Strigiformes and Apodiformes (145 samples each), and Charadriiformes (58), which, combined, covered 91.4% of all collected birds. The mean prevalence for USUV and WNV was 4.5% and 6.5% respectively, while the co-infection rate reached 1.5%.

At the Taxonomic Order level, Columbiformes appeared to have the highest USUV prevalence (28/345; 8%), followed by Anseriformes (1/14; 7%), Ciconiformes (2/32; 6%), and Galliformes (1/16; 6%). Among the most frequently found-dead species (at least covering 2% of the total tested birds), Common wood pigeon (*Columba palumbus)* showed the highest prevalence to each single virus (19.3%) and the highest rate of animals co-infected by both viruses (7.9%).

Galliformes (2/16) and Strigiformes (19/145) scored the highest prevalence for WNV (13%), followed by Columbiformes (35/345; 10%) and Charadriformes (4/58; 7%). The order Passeriformes showed an average prevalence of 5% (40/794) for each virus, although European blackbird (*Turdus merula),* the most prevalent species among Passeriformes, reached a 13% USUV prevalence, 4.3% to WNV, and five birds were co-infected by both viruses. Other notable infected species that were less frequently recovered were greenfinch (*Carduelis chloris*), scops owl (*Otus scops*), house martin (*Delichon urbica*), house sparrow (*Passer domesticus*), and pheasant (*Phasianus colchicus*), which reported high WNV prevalence: 33.3%, 21.4%, 20.0%, 14.7%, and 12.5%, respectively.

We recorded the highest number of PCR positive birds between July and September, with a peak in the second fortnight of August (Figure 1).

Four strains of USUV were obtained from three blackbirds (*Turdus merula*) sampled in the years 2015–2017 and one common wood pigeon (*Columba palumbus*) collected in 2019. All these birds were PCR negative for WNV. No isolation of WNV was obtained from the birds included in the study.

We obtained 30 partial NS5 gene sequences for WNV from 14 different bird species collected between 2015 and 2019; the more represented species were magpies (6 sequences), little owls (4 sequences), and blackbirds (4 sequences). Four synonymous mutations were registered in these sequences; the overall p-distance among these sequences was 0.004 (with a maximum of 0.014). Among the sequences with a complete coverage, the consensus have the highest identity (99%) with two complete genomes of strains isolated from a man in Veneto region (MN939564, MN939563) in 2016 and 2018.

In the 29 partial gene E sequences of WNV, six mutations were found, four of which were not synonymous. The overall p-distance among these sequences was 0.002 (with a maximum of 0.013). The majority of the sequences carry an A in position 289. The presence of this base is characteristic of clade B of the Italian WNV lineage 2 (WNV-2) sequences, following the classification proposed by Veo et al. [17]. The sequences closest to our consensus (one different base, 99.7% identity) deposited in GenBank are complete genomes, isolated from mosquitoes sampled in Emilia-Romagna in 2013 (KU573083, KU573080) and Austria in 2014 (KP109692), and one from a blood donor of Austria in 2015 (MF984340).

We obtained 19 partial NS5 gene sequences for USUV from five different bird species, of which 12 were from blackbirds, collected between 2015 and 2019. Four synonymous point mutations were present; the more common, an A > G mutation in position 120, was recorded in ten sequences. This mutation was first detected in 2018 and is not yet present in GenBank. The overall p-distance among these sequences was 0.005 (with a maximum of 0.014). The most similar sequences to our consensus in GenBank (99.59% identity, corresponding to the mutation above described) were related to mosquitoes collected in Emilia-Romagna between 2009 and 2010 (HM138707, HM138708, HM138710, HM138714, HM138718, JF834546, JF834549, JF834550, JF834556, JF834560, JF834591).

We analyzed 67 partial E gene sequences for USUV obtained from 21 different bird species collected between 2015 and 2019. The most represented species were blackbirds (21 sequences), magpies (9 sequences), jays (6 sequences), and wood pigeons (3 sequences). The overall p-distance among these sequences was 0.003 (with a maximum of 0.028) and we registered 25 mutations, out of which 5 were non synonymous. Among the sequences present in GenBank, the most similar ones, with a 100% identity, originated from mosquitoes (2010) (JF834599, JF834604, JF834606, JF834616, JF834623, JF834626, JF834673) and from one human case of neurological disease (2009) in Emilia-Romagna (JF826447). These findings confirm that the USUV strain circulating in the province of Ferrara belongs to the Europe 2 (EU2) clade, a clade detected in North Italy since 2009 [5].

## 4. Discussion

Usutu and West Nile viruses are Flaviviruses maintained in the environment through an enzootic cycle involving mosquitoes and wild birds. In temperate climate areas, the seasonal pattern of the infection shows an increase during warmer months, owing to the involvement of mosquitoes in the transmission cycle of the virus.

As USUV and WNV are zoonotic agents, it is pivotal to run and constantly improve surveillance programs for early detection of their circulation. The integrated WNV surveillance program in place in the province of Ferrara since 2009 includes the virological monitoring of the mosquito vectors (mainly *Culex pipiens*), the surveillance on birds shot (active surveillance) and found dead (passive surveillance), and the testing of all cases of neurological disease in horses and of all viral meningoencephalitis cases in humans [7].

Compared with active surveillance (i.e., entomological or on animal sentinels), passive surveillance systems (i.e., on dead-found birds or on diseased horses) are simpler and cheaper to implement [18], but the challenge is that they should be capable of early detection of the virus before human cases occur. A limitation of passive surveillance in birds is that not all birds (species) would die from infection. Therefore, unless the bird dies from infection or from another cause, those cases are not captured. Thus, in order to increase the sensibility of passive surveillance, a high number of subjects have to be examined. Moreover, the time elapsed between the death and the delivery of the bird carcasses to the laboratory should be reduced as much as possible.

The province of Ferrara includes one of the largest wetlands of Europe where mosquitoes and wild birds coexist, and where USUV and WNV have both been endemic for several years [3]. Thanks to these characteristics, the province of Ferrara offers the best scenario to study which bird species are the most suitable target for passive surveillance monitoring.

Our results show a wide abundance and variety of wild birds, mainly belonging to six orders, with Passeriformes being the most represented. The majority of found-dead-birds were recovered between May and August. This is probably because of the higher collection of yearlings in April and in May, because of their early leaving of the nest or attacks on these birds by predators. During summer months, birds are more often collected because common people, spending much time outdoor, have higher chances of finding sick or injured animals.

Throughout the study period, the average USUV and WNV prevalence detected by PCR was 4.5% and 6.5%, respectively, confirming their endemic presence. Detected sequences belong to WNV lineage 2 and USUV clade EU2. For both viruses, the low mutations number in sequences suggests a recurrent circulation of the same strain rather than the reintroduction of new ones.

Birds were infected mainly between July and September, with a peak in the second fortnight of August, consistent with the peak of mosquitoes’ positivity already described for this wetlands area [4].

This research did not investigate the causes of death in birds, so it is not possible to determine if they died from the virus or from other causes while they were infected. Scientific literature shows that there are significant differences among species, but a high percentage of birds die within 7 days after experimental infection with WNV or USUV, and that viruses can be found about two weeks after infection (Table 2), although persistent infection from WNV could not be excluded [19].

Between 2015 and 2019, on average, more than 350 bird were examined per year. This high number of tests increases the sensibility of passive surveillance, which is thus able to detect the virus every single year, often before the first human case occurs (Table 3). In the considered period, 19 WNV neurological disease cases were reported in humans and no USUV encephalitis cases. Moreover, no neurological disease was recorded in horses, probably because of the high immunity of this population owing to natural infection or vaccination.

About 1.5% of the birds tested PCR positive for both viruses. PCR is considered to be highly specific, but cross reactions can also occur because of the high similarity among these viruses [34]. In the considered period, we isolated WNV in a dunnock (*Prunella modularis*) that tested negative for USUV. This bird was not included in this research because it came from another province (Modena), although it was collected in the same recovery center.

In our study, the Collared dove and the Common wood pigeon (Columbiformes) scored the highest mean prevalence for USUV. Similarly, the Ring-necked pheasant (Galliformes) and the Little owl (Strigiformes) scored the highest prevalence for WNV. The Eurasian blackbird (*T. merula*) is usually referred to as the most USUV-susceptible species [35,36]. In our study, the blackbird seemed to be particularly prone to co-infection by both viruses, suggesting it could represent a good proxy for the early detection of these Flaviviruses. Considering the importance of surveillance programs for the early detection of USUV and WNV, our data support the adoption of passive surveillance mainly focused in the summer period (from July to mid-September), and directed towards the species of wild birds more prone to die as a result of infection by these viruses, such as Passeriformes, Strigiformes, and Columbiformes.

The involvement of the staff working in the wildlife recovery centers makes it easier to achieve the number of samples required to give sufficient sensitivity to the surveillance system.

## Figures and Tables

**Figure 1 viruses-13-01367-f001:**
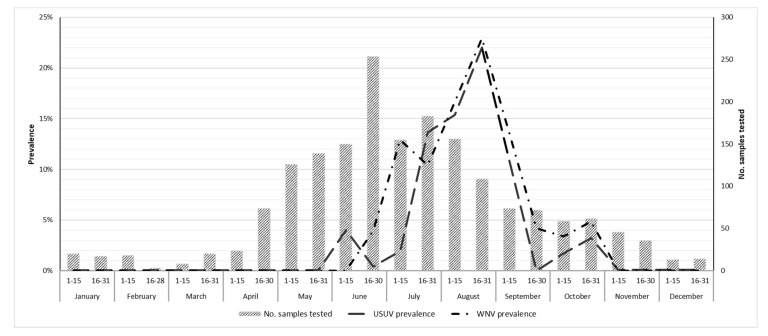
Number of birds tested for Usutu (USUV) and West Nile (WNV) viruses and fortnightly prevalence, province of Ferrara, Italy, 2015–2019.

**Table 1 viruses-13-01367-t001:** Virological findings in wild birds collected in the province of Ferrara between 2015 and 2019. USUV, Usutu virus; WNV, West Nile virus.

Order	Common Name	Scientific Name	Migration Pattern	No. Tested Birds	No. USUV Positive (%)	No. WNV Positive (%)	No. Co-Infected (%)
Accipitriformes	Common Buzzard	*Buteo buteo*	R,P,S	36	0 (0%)	0 (0%)	0 (0%)
	Eurasian Sparrowhawk	*Accipiter nisus*	L	7	0 (0%)	0 (0%)	0 (0%)
	Western Marsh Harrier	*Circus aeruginosus*	L	3	0 (0%)	0 (0%)	0 (0%)
	Subtotal			47	0 (0%)	0 (0%)	0 (0%)
Anseriformes	Mallard	*Anas platyrhynchos*	R,P,S	7	1 (14.3%)	0 (0%)	0 (0%)
**	Other 4 species *	**		7	0 (0%)	0 (0%)	0 (0%)
	Subtotal			14	1 (7.1%)	0 (0%)	0 (0%)
Apodiformes	Common Swift	*Apus apus*	L	145	7 (4.8%)	7 (4.8%)	3 (2.1%)
Charadriiformes	European Herring Gull	*Larus argentatus*	R,P,S,L	46	3 (6.5%)	3 (6.5%)	1 (2.2%)
	Common Tern	*Sterna hirundo*	L	1	0 (0%)	1 (100%)	0 (0%)
**	Other 4 species *	**		11	0 (0%)	0 (0%)	0 (0%)
	Subtotal			58	3 (5.2%)	3 (5.2%)	1 (1.7%)
Ciconiiformes	Cattle Egret	*Bubulcus ibis*	L	7	1 (14.3%)	0 (0%)	0 (0%)
	Purple Heron	*Ardea purpurea*	L	5	1 (20%)	0 (0%)	0 (0%)
**	Other 5 species *	**		20	0 (0%)	0 (0%)	0 (0%)
	Subtotal			32	2 (6.2%)	0 (0%)	0 (0%)
Columbiformes	Collared Dove	*Streptopelia decaocto*	R	257	11 (4.3%)	18 (7%)	3 (1.2%)
	Common Wood Pigeon	*Columba palumbus*	R,P,S	88	17 (19.3%)	17 (19.3%)	7 (7.9%)
	Subtotal			345	27 (7.8%)	35 (10.1%)	10 (2.9%)
Coraciiformes	European Bee-eater	*Merops apiaster*	L	7	0 (0%)	0 (0%)	0 (0%)
Cuculiformes	Cukoo	*Cuculus canorus*	L	4	0 (0%)	0 (0%)	0 (0%)
Falconiformes	Common Kestrel	*Falco tinnunculus*	R,P,S	146	3 (2.1%	9 (6.2%)	2 (1.4%)
	Eurasian Hobby	*Falco subbuteo*	L	3	0 (0%)	1 (33.3%)	0 (0%)
	Red-footed Falcon	*Falco vespertinus*	M	1	0 (0%)	0 (0%)	0 (0%)
	Subtotal			150	3 (2.0%)	10 (6.7%)	2 (1.3%)
Galliformes	Ring-necked Pheasant	*Phasianus colchicus*	R	16	1 (6.3%)	2 (12.5%)	1 (6.3%)
Gruiformes	Common Moorhen	*Gallinula chloropus*	R	25	1 (4%)	0 (0%)	0 (0%)
	Water-cheeked Rail	*Rallus aquaticus*	M	1	0 (0%)	0 (0%)	0 (0%)
	Subtotal			26	1 (3.8%)	0 (0%)	0 (0%)
Passeriformes	European Blackbird	*Turdus merula*	R,P	184	24 (13%)	8 (4.3%)	5 (2.7%)
	Eurasian Magpie	*Pica pica*	R	166	4 (2.4%)	2 (1.2%)	0 (0%)
	Eurasian Jay	*Garrulus glandarius*	R,P	81	2 (2.5%)	3 (3.7%)	0 (0%)
	Common Starling	*Sturnus vulgaris*	R,P,S	64	1 (1.6%)	1 (1.6%)	0 (0%)
	Barn Swallow	*Hirundo rustica*	L	43	2 (4.7%)	5 (11.6%)	0 (0%)
	Great Tit	*Parus major*	R,S	40	2 (5%)	2 (5%)	2 (5%)
	Hooded Crow	*Corvus cornix*	R	40	0 (0%)	3 (7.5%)	0 (0%)
	House Sparrow	*Passer domesticus*	R	34	1 (2.9%)	5 (14.7%)	0 (0%)
	Common House Martin	*Delichon urbicum*	L	15	1 (6.7%)	3 (20%)	1 (6.7%)
	European Goldfinch	*Carduelis carduelis*	L	14	2 (14.3%)	1 (7.1%)	0 (0%)
	European Greenfinch	*Carduelis chloris*	L	12	0 (0%)	4 (33.3%)	0 (0%)
	Eurasian Reed Warbler	*Acrocephalus scirpaceus*	L	6	1 (16.7%)	0 (0%)	0 (0%)
	Eurasian Golden Oriole	*Oriolus oriolus*	L	4	1 (25%)	0 (0%)	0 (0%)
	Blue Tit	*Cyanistes caeruleus*	L	3	1 (33.3%)	1 (33.3%)	1 (33.3%)
	European Serin	*Serinus serinus*	L	3	1 (33.3%)	0 (0%)	0 (0%)
	European Pied Flycatcher	*Ficedula hypoleuca*	L	2	0 (0%)	1 (50%)	0 (0%)
	Common Redstart	*Phoenicurus phoenicurus*	L	1	0 (0%)	1 (100%)	0 (0%)
**	Other 22 species *	**		82	0 (0%)	0 (0%)	0 (0%)
	Subtotal			794	43 (5.4%)	40 (5.0%)	9 (1.1%)
Pelecaniformes	Great Cormorant	*Phalacrocorax carbo*	L	5	0 (0%)	0 (0%)	0 (0%)
Phoenicopteriformes	Flamingo	*Phoenicopterus roseus*	L	3	0 (0%)	0 (0%)	0 (0%)
Piciformes	European Green Woodpecker	*Picus viridis*	R,P	48	0 (0%)	1 (2.1%)	0 (0%)
	Great Spotted Woodpecker	*Dendrocopos Major*	R,M	2	0 (0%)	0 (0%)	0 (0%)
	Eurasian Wryneck	Jynx torquilla	R,M	1	0 (0%)	0 (0%)	0 (0%)
	Subtotal			51	0 (0%)	1 (2.0%)	0 (0%)
Podicipediformes	Great Crested Grebe	*Podiceps cristatus*	L	1	0 (0%)	0 (0%)	0 (0%)
Strigiformes	Barn Owl	*Tyto alba*	R	11	0 (0%)	1 (9.1%)	0 (0%)
	Little Owl	*Athene noctua*	R	84	3 (3.6%)	13 (15.5%)	1 (1.2%)
	Long-eared Owl	*Asio otus*	R,P,S	35	0 (0%)	2 (5.7%)	0 (0%)
	Eurasian scops Owl	*Otus scops*	L	14	1 (7.1%)	3 (21.4%)	1 (7.1%)
	Tawny Owl	*Strix aluco*	R	1	1 (100%)	0 (0%)	0 (0%)
	Subtotal			145	5 (3.4%)	20 (13.8)	2 (1.4%)
Total	**	**	**	1842	94 (4.5%)	118 (6.5%)	28 (1.5%)

Note: R = resident; P = partial migrant; S = short distance migrant; L = long distance migrant. * = the complete list of the bird species is reported in Appendix A.

**Table 2 viruses-13-01367-t002:** Review of experimental infection with West Nile or Usutu viruses in birds.

Order	Common Name	Scientific Name	Virus Used	Infection Route	Dose	Died/Infected (%)	Mean No. Days to Death (Range)	Maximum Viral Persistence in Organs	Ref.
Accipitriformes	American Kestrel	*Falco sparverius*	WNV 1	MB, OS	Various *	0/5 (0%)	-	11 d.p.i.	[20]
Anseriformes	Domestic Goose	*Anser anser domesticus*	USUV 1	IM	5 × 10^4^ TCID_50_	5/11 (45%)	4.6 (2–16)	16 d.p.i.	[21]
Charadriformes	Killdeer	*Charadrius vociferus*	WNV 1	MB, OS	5–15 mosquitoes *	0/2 (0%)	-	10 d.p.i.	[20]
Columbiformes	Mourning Dove	*Zenaida macroura*	WNV 1	MB, OS	Various *	0/6 (0%)	-	11 d.p.i.	[20]
Falconiformes	Gyr-Saker hibrid falcon	*Falco rusticolus x Falco cherrug*	WNV 1	SC	10^4^ TCID_50_	0/6 (0%)	-	14 d.p.i.	[22]
	Gyrfalcon and hibrid falcon	*Falco rusticolus*	WNV 1	SC	500 TCID_50_, 10^4^ TCID_50_, 10^6^ TCID_50_	4/12 (30%)	9.75 (8–12)	21 d.p.i.	[23]
Galliformes	Domestic Chicken	*Gallus domesticus*	USUV 1	IV	10^3^ TCID_50_	0/18 (0%)	-	7 d.p.i.	[24]
	Japanese Quail	*Coturnix japonica*	WNV 1	MB, OS	Various *	0/6 (0%)	-	14 d.p.i.	[20]
	Red-legged Partridge	*Alectoris rufa*	WNV 1	SC	10^7^ PFU	3/13 (23%)	5.7 (2–8)	10 d.p.i.	[25]
Passeriformes	Domestic Canary	*Serinus canaria domestica*	USUV 2	IP	10^3^ TCID_50_; 10^6^ TCID_50_	3/10 (30%)	7 (5–9)	NR	[26]
	Domestic Canary	*Serinus canaria domestica*	WNV	SC	10^5^ PFU,10^2^ PFU,10 PFU	0/23 (0%)	-	5 d.p.i.	[27]
	House Sparrow	*Passer domesticus*	WNV 1	MB, OS	Various *	3/15 (20%)	4.7 (3–6)	10 d.p.i.	[20]
	House Sparrow	*Passer domesticus*	WNV 1	SC	10^5^ PFU	5/24 (21%)	5.6 (4–7)	21 d.p.i.	[28]
	House Sparrow	*Passer domesticus*	WNV 1	SC	1000–4000 PFU	19/150 (13%)	(8–354)	65 d.p.i.	[19]
	Red-winged Blackbird	*Agelaius phoeniceus*	WNV 1	MB, OS	5–15 inf. Mosquitoes *	0/4 (0%)	-	11 d.p.i.	[20]
	Common Grackle	*Quiscalus quiscula*	WNV 1	MB, OS	Various *	2/12 (17%)	4.5 (4–5)	10 d.p.i.	[20]
	Fish Crow	*Corvus ossifragus*	WNV 1	MB, OS	Various *	5/20 (25%)	9.6 (6–13)	9 d.p.i.	[20]
	Blue Jay	*Cyanocitta cristata*	WNV 1	MB, OS	Various *	3/6 (50%)	4.7 (4–5)	9 d.p.i.	[20]
	Eurasian Magpie	*Pica pica*	WNV 1,2	SC	5000 PFU	13/19 (68%)	6.6 (6–8)	17 d.p.i.	[29]
	Carrion Crows	*Corvus corone*	WNV 1,2	NR	2000 TCID_50_	22/30 (73%)	7.3 (6–9)	14 d.p.i.	[30]
	Carrion Crows	*Corvus corone*	WNV 1	SC	1500 PFU	8/12 (67%)	6.9 (5–10)	10 d.p.i.	[31]
	European Jackdaws	*Corvus monedula*	WNV 1,2	NR	2000 TCID_50_	11/26 (42%)	7.3 (5–9)	14 d.p.i.	[32]
Psittaciformes	Budgerigar	*Melopsittacus undulatus*	WNV 1	MB, OS	Various *	0/6 (0%)	-	13 d.p.i.	[20]
Strigiformes	Eastern Screech Owl	*Megascops asio*	WNV 1	SC, OS	1000–2000 PFU, 1 inf. mouse	2/9 (22%)	8.5 (8–9)	14 d.p.i.	[33]

Note: NR = not reported, SC = subcutaneous, IM = intramuscular, IV = intravenous, MB = infected mosquito bites, IP = intraperitoneal, d.p.i. = days post infection; * birds exposed to 5–15 WNV-infected mosquitoes, or infected with a 200 μL suspension of WNV in oral cavity, or one dead infected mosquito in oral cavity, or one dead infected mouse in oral cavity.

**Table 3 viruses-13-01367-t003:** Timing of detection of West Nile virus infection in birds, equids, and humans, province of Ferrara, 2015–2019.

	Birds		Equids		Human Beings	
Year	Total WNV Positive Birds	Date of Death of the First Positive Bird	Total Cases of WNV Disease	Date of Onset of Signs of the First Case	Total Cases of WNV Neurological Disease	Date of Onset of Signs of The First Case
2015	14	20 July 2015	0		1	13 August 2015
2016	6	6 August 2016	0		2	18 July 2016
2017	18	17 August 2017	0		2	4 August 2017
2018	45	4 June 2018	0		14	18 July 2018
2019	11	22 July 2019	0		0	

## Data Availability

Data on human cases of West Nile neurological disease reported in Table 3 are available on the National Bulletins of West Nile and Usutu viruses human infections (available online: https://www.epicentro.iss.it/westnile/bollettino).

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
