# Peer review of "West Nile and Usutu Viruses’ Surveillance in Birds of the Province of Ferrara, Italy, from 2015 to 2019"

_viruses, 2021, doi:10.3390/v13071367_

Round 1

Reviewer 1 Report

All previous comments provided by this reviewer were addressed by the authors in this revised version of the manuscript.

Author Response

We warmly thank the reviewer for his/her time and effort in reading this new version of the manuscript.

Reviewer 2 Report

The article reads very well and provides information that contribute to strengthening surveillance of WNV and USUV in Europe.  The authors have clearly made an effort in addressing my previous comments.

I only have few comments to be considered by the authors:

  • Line 78: I suggest removing the word “briefly”
  • Table 1: Some of the species are not mentioned (e.g. “Other 4 species”). It is unclear why it is the case, except that there were no positive findings is these species. Negative findings are also relevant, so the authors (if the information is available) could put the list of all these unspecified species in an annex or simply list them below the table.
  • Line 161: “and from one case of neurological disease”; the authors should specify whether this is a human or equine case.
  • Line 167: “probably due to the involvement of mosquitoes”; The authors should consider removing the word “probably” from this sentence.
  • Line 176: “passive surveillance systems (i.e. on dead-found birds or on diseased horses) are probably simpler and cheaper to implement”; The authors should consider removing the word “probably” from this sentence.
  • Line 175-180: Another limitation of passive surveillance in birds is that not all birds (species) would die from infection. Therefore, unless the bird dies from infection or from another cause, those infections are not captured. The authors could consider adding this limitation.

Author Response

We warmly thank the reviewer for his/her time and effort in reading this new version of the manuscript.

We have considered all the suggestions:

  • Line 78: I suggest removing the word “briefly”

R: “briefly” was deleted as suggested

  • Table 1: Some of the species are not mentioned (e.g. “Other 4 species”). It is unclear why it is the case, except that there were no positive findings is these species. Negative findings are also relevant, so the authors (if the information is available) could put the list of all these unspecified species in an annex or simply list them below the table.

R: We tried to list all the species in one table buti it resulted to be too big for the journal standards. We also tried to list them below the table but this resulted in an excessively long list which, in our opinion, would impact the readability of the manuscript. Thus we decided for a different approach: we left the table unchanged reporting the more significant species, while we compiled a full list of all the species in an Excel file that will be sent as “Supplementary material”.

  • Line 161: “and from one case of neurological disease”; the authors should specify whether this is a human or equine case.

R: we have specified that it was a human case of disease.

  • Line 167: “probably due to the involvement of mosquitoes”; The authors should consider removing the word “probably” from this sentence.

R: The word “probably” was removed as suggested

  • Line 176: “passive surveillance systems (i.e. on dead-found birds or on diseased horses) are probably simpler and cheaper to implement”; The authors should consider removing the word “probably” from this sentence.

R: The word “probably” was removed as suggested

  • Line 175-180: Another limitation of passive surveillance in birds is that not all birds (species) would die from infection. Therefore, unless the bird dies from infection or from another cause, those infections are not captured. The authors could consider adding this limitation.

R: We agree with the reviewer’s comment. We added a sentence at lines 178-180, and modified the text at line 240.